# Orthorexia nervosa and social media: A mixed-methods scoping review using a systematic methodology

Emmanuelle Awad [1,2*], Jessica M. Alleva[1], Celine El Khoury[3], Nour Chamma[3], Carolien Martijn[1], Rana Rizk[3,4]

1 Faculty of Psychology and Neuroscience, Department of Clinical Psychological Science, Maastricht University, The Netherlands, 2 Department of Psychology and Education, Lebanese American University, Lebanon, 3 School of Arts and Sciences, Department of Nutrition and Food Science, Lebanese American University, Byblos, Lebanon, 4 INSPECT-LB (Institut National de Santé Publique, d'Épidémiologie Clinique et de Toxicologie-Liban), Lebanon

☯ These authors contributed equally to this work.
* emmacawad@gmail.com

## Abstract

### Introduction

Orthorexia Nervosa (ON) is defined as a preoccupation with abiding by a self-perceived healthy diet resulting in pathological thought, behavior and emotion. Social media platforms may add to this preoccupation with healthy eating by offering accessible exposure to food and dieting information. Yet, the precise nature of the association between ON and exposure to social media remains underexplored. The aim is to explore the existing literature on the relationship between social media and ON in a scoping search and to synthesize the findings.

### Methods

We searched PubMed, Web of Science, PsycInfo (Ovid), ProQuest, and Embase (Elsevier). As for the grey literature, we searched ProQuest Dissertations and Open Access Theses and Dissertations. The first literature search was conducted on August 7, 2023 then a literature update on September 18, 2024.

### Results

After the identification of studies and data extraction, authors assessed the methodological quality of included studies. The characteristics and findings from the studies were narratively synthesized, with a focus on the platform, duration, and content of social media use. A total of 31 studies were identified between 2017 and 2024, which were predominantly cross-sectional and focused on Westernized populations. The results have shown a bidirectional relationship between ON and social media, influenced by the characteristics of the platform (e.g., image-based), duration of use

**Data availability statement:** All relevant data are within the manuscript and its Supporting Information files.

**Funding:** The author(s) received no specific funding for this work.

**Competing interests:** The authors have declared that no competing interests exist.

(e.g., longer use), content themes (e.g., diet and fitness-related), and individual-level factors (e.g., limited health literacy, young adulthood and adolescence, and body dissatisfaction).

## Conclusion

The main research gaps found were the absence of experimental studies, lack of studies in non-Westernized populations, restrictive samples, and lack of investigations about developmental pathways. Longitudinal, experimental and qualitative studies in future research are warranted to advance knowledge. Systematic review registration: Open Science Framework (OSF) Registries (DOI https://osf.io/vrqwt).

---

## Introduction

Orthorexia Nervosa (ON) is characterized by an obsession with healthy eating and a fixation on eating 'proper' food, with a prevalence of 6.9% in the general population and 35-57.8% in high-risk groups.[1] ON poses significant risks to individuals' physical, psychological, and social well-being.[2] Individuals who exhibit ON might be anxious when presented with food that is perceived as unhealthy, and may experience negative emotions such as guilt.[3] The term "Orthorexia Nervosa" was first introduced by Bratman to reflect a pathological adherence to a diet for health motives that produces maladaptive eating behaviors and negative psychological outcomes.[4] ON has also been described as a quest towards perfect "dietary purity." [5]

Despite the lack of consensus on its diagnostic criteria, [6] some common elements can be identified among the different proposed definitions of ON. First, ON involves extensive forethought and application of rigid dietary regulations. Subsequently, this strict self-monitoring of diet contributes to harmful consequences that can extend to multiple domains including physical (e.g., malnutrition and potential nutritional deficiencies), psychological (e.g., obsessive thinking and emotional distress), and social (e.g., isolation) domains.[3] Based on these previous findings, the current study will define ON as a preoccupation with abiding by a self-perceived healthy diet resulting in pathological thought, behavior and emotion.

While not officially recognized as a disorder by psychiatric boards, the need for further scientific analysis into ON is evident.[7,8] While sharing traits with Anorexia Nervosa (AN) and Obsessive-Compulsive Disorder (OCD), ON is distinct in its motivation, emphasizing a desire for health rather than weight control or fear of higher body weight.[9,10] The study of Koven and Abry [9] showed that orthorexic individuals may experience dysfunction in domains such as attention and working memory, independently of AN and OCD scores.[9] This has led to a call for new assessment tools to better understand ON.[7]

Instagram, and social media in general, have been linked to mental health concerns. For example, a study by Lin et al.[11] indicated a strong and significant relationship between social media use and depression, highlighting the need for further research to understand the nuances of this association. Turner and Lefevre [12]

exploration of various social media channels, including Instagram, Facebook, Twitter, Pinterest, Google +, Tumblr, and LinkedIn, revealed a distinctive positive relationship between Instagram use and ON. Their results further underscore the need for an understanding of the relationship between different social media platforms and the development and progression of eating disorders, which could facilitate future research and interventions in this evolving landscape.[12] Despite the growing concern over the impact of social media on mental health, the specific pathways underlying the relationship between social media use and ON remain unclear.[13,14]

### Rationale and aim

Though not officially considered a psychiatric disorder, the unofficial recognition of ON has drawn considerable attention. The existing literature underlines important links between ON and social media use, including its role in worsening disordered eating patterns, promoting unhealthy comparisons, and influencing dietary behaviors [15]. Social media platforms play a notable role in the onset of ON [16], yet the specific mechanisms and patterns remain underexplored, leaving significant gaps in understanding requiring further exploration. First, social media platforms offer distinct immersive and interactive experiences for users. A scoping review can help to outline how different social media platforms can variously contribute to ON. Second, not only is social media evolving rapidly in terms of features, but also in content trends. This reinforces the value of organizing research about the content displayed on social media in order to explore the current dynamics possibly influencing the relationship between ON and social media. Additionally, the literature offers a significant number of studies about ON and social media, but specific mechanisms and pathways remain underexplored. Given that social media differs in terms of platform used, time spent and content consumed, a synthesis of findings would clarify the relationship between ON and social media. Therefore, the current study will explore the existing literature on the relationship between social media and ON in a scoping search and synthesis.

### Research question

What is the relationship between ON and social media use, and what individual-level factors are associated with this relationship?

## Methods

### Review design

This scoping review followed a protocol that was registered with the Open Science Framework (OSF) Registries (DOI https://osf.io/vrqwt). We followed the Preferred Reporting Items for Systematic reviews and Meta-Analyses (PRISMA) statement [17] and its literature search extension (PRISMA-S) [18], along with the extension for scoping reviews (PRISMA-ScR) [19] to report on the current scoping review and its search strategy. No ethical approval was required. Given that the screened and included studies were very heterogeneous in terms of characteristics such as study design, populations, and scales used, we determined that a meta-analysis of the data was not appropriate and restricted our synthesis of the data to a narrative review.

### Criteria for study inclusion

The Problem/Population, Intervention/Exposure, Comparison, and Outcome (PICO/PECO) format was used to establish the inclusion criteria. Original research including quantitative, qualitative, mixed-methods, and observational studies discussing ON and social media were included. Studies meeting these criteria involving samples of all age groups and all genders were included. Excluded were books, book chapters, editorials, case studies, case series, reviews, systematic reviews, blogs, studies published as posters or in conference proceedings and commentaries. We did not include any publication language nor publication date restrictions.

## Search strategy

We searched PubMed, Web of Science, PsycInfo (Ovid), ProQuest, and Embase (Elsevier). As for the grey literature, we searched ProQuest Dissertations and Open Access Theses and Dissertations (OATD). The search on each database/interface included two concepts: ON and social media, using controlled vocabulary, such as Medical Subject Headings (MeSH) in combination with free-text words. We also contacted key scholars in the field to ask for unpublished studies, and we distributed calls for research via listservs on related research topics (i.e., Michael Levine's Eating Disorders Prevention/Sociocultural Factors Newsletter). Following that, we conducted a citation search based on the references of included studies. The search strategy was validated by a medical information specialist. The first literature search was conducted on August 7, 2023 by two authors (EA/RR). A literature update was also conducted on September 18, 2024.

## Study selection

The screening process was done independently and in duplicate by two pairs of researchers (EA/CEK and EA/NC) using EndNote X8. It that consisted of assessing titles and/or abstracts of studies that were retrieved by the search. Second, the full texts of published and unpublished studies that were eligible were retrieved and evaluated independently and in duplicate by two pairs of researchers (EA/CEK and EA/NC). Disagreements relevant to titles and/or abstracts screening and full texts screening were discussed between pairs of authors. If an agreement was not reached, a third reviewer was consulted (RR). Calibration exercises preceded the assessment of titles and/or abstracts, and full texts screening to enhance inter-rater agreement.

## Data extraction and quality assessment

Following the identification of eligible studies, the data were extracted by two pairs of authors (EA/CEK and EA/NC) independently and in duplicate. The authors used a data extraction form that covered information about the author-recorded study pre-registration, methodology, results, limitations, and conclusion. Then, the same pairs of authors (EA/CEK and EA/NC) used the Mixed Methods Appraisal Tool (MMAT) Version 18 [20] to assess the methodological quality of included studies. The MMAT includes two screening questions to confirm that the study is an empirical study, followed by five questions to critically appraise the studies depending on the design (qualitative, quantitative randomized controlled trials, quantitative non-randomized, quantitative descriptive, and mixed-methods). A calibration exercise was done before the data extraction and quality assessment. Disagreements were resolved through discussions, if not resolved, a third reviewer was consulted (RR).

## Data synthesis

We narratively synthesized the findings from the studies. We focused on the platform, duration, and content of social media use. We meta-analyzed the findings if the populations under study, methods followed, and results were similar enough to allow pooling, using the Review Manager (RevMan) Software version 5.4 (The Cochrane Collaboration, The Nordic Cochrane Centre). For a meta-analysis, the minimum number of studies needed was two studies.[21] A *p*-value of less than 0.05 was considered statistically significant.

## Results

### Characteristics of included studies

This review included 31 studies on ON and social media use (Fig 1). The studies covered a variety of countries including Türkiye, the USA and Western-European countries of which Germany, Italy, and the UK were particularly well represented. Sample sizes varied from small qualitative cohorts such as in the study by Cheshire, Berry [22] with 9 participants to large cross sectional surveys such as in the study by Asil, Yılmaz [23] with over 2,500 participants. Mean ages of the

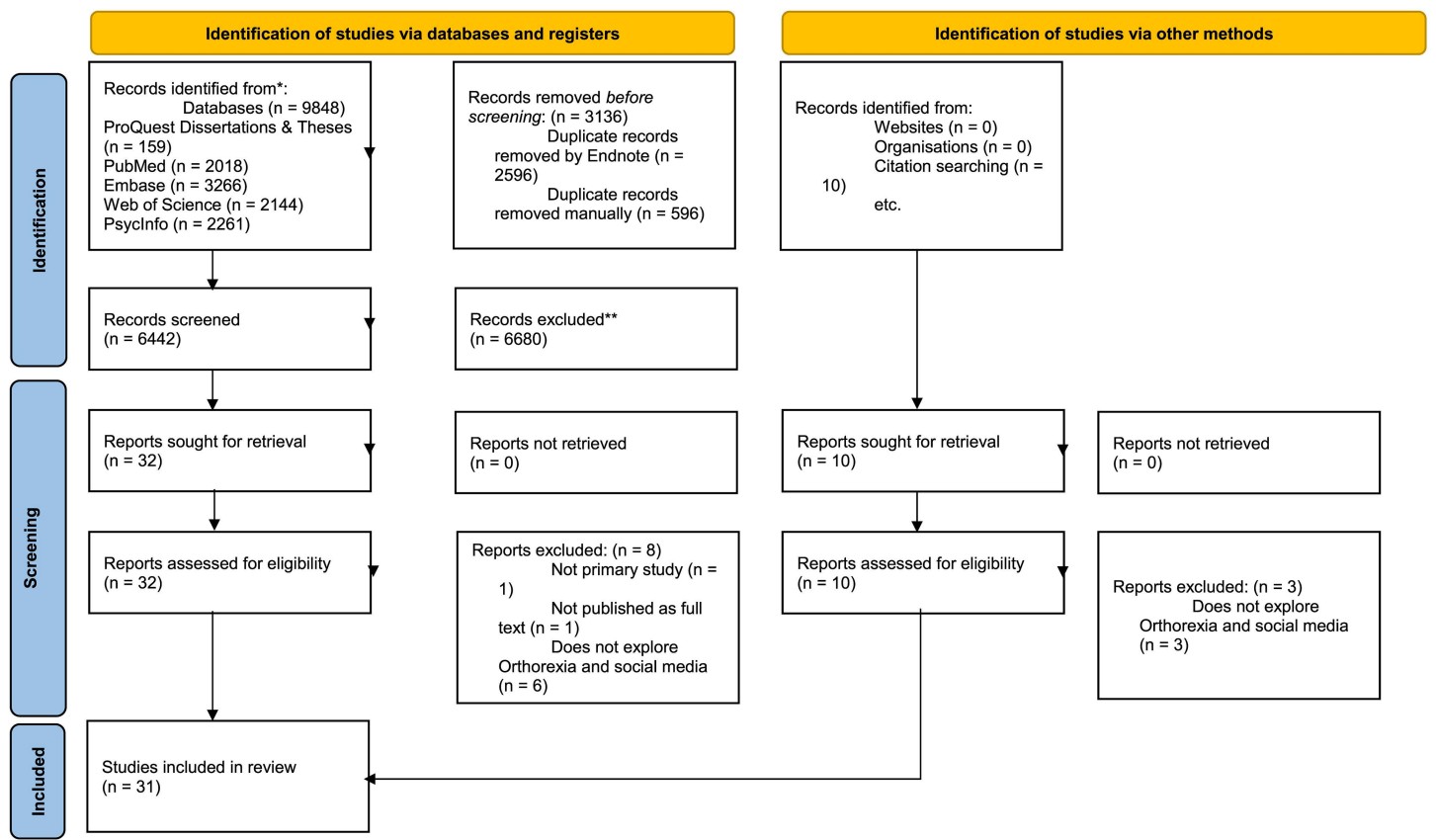

**Fig 1. PRISMA flowchart for literature search.** The PRISMA flowchart illustrates the literature search and selection process. The flowchart includes multiple stages: identification, screening, eligibility, and inclusion. Each stage shows the number of records retrieved, duplicates removed, records screened, full-text articles assessed, exclusions with reasons, and final studies included in the review. The flowchart visually represents the filtering of studies from an initial database search to the final selection. *Consider, if feasible to do so, reporting the number of records identified from each database or register searched (rather than the total number across all databases/registers). **If automation tools were used, indicate how many records were excluded by a human and how many were excluded by automation tools. (Adapted from Page MJ, McKenzie JE, Bossuyt PM, Boutron I, Hoffmann TC, Mulrow CD, et al. The PRISMA 2020 statement: an updated guideline for reporting systematic reviews. BMJ 2021;372:n71. https://doi.org/10.1136/bmj.n71).

participants in most studies were in the early twenties. Gender distribution was varied, with women being the most prevalent in 16 studies, and the studies of Cheshire, Berry [22] Gobin, Mills [24] Issa [25] Norton [26] Piko, Kulmán [27] and Turner and Lefevre [12] including female participants only. Meanwhile, the studies by Yilmaz and Unal [28] and Yılmazel [29] had a majority of male participants, and the study of Karniej, Pérez [30] had only male participants. Several studies of ON and social media use were performed, using a variety of tools to assess ON including standardized questionnaires such as ORTO-15 [31] and Teruel Orthorexia Scale (TOS) [32]. In addition, several studies used self-constructed surveys to capture specific social media behaviors related to their research questions (Table 1). In addition, Table 1 covers the overall quality rating of each study on a scale from 1 to 7, with 7 being the best index of quality. This rating was done based on the MMAT Version 18 [20].

## Methodology of included studies

A total of 22 the studies were quantitative, cross-sectional designs to determine the prevalence of ON symptoms and their associations with social media use. Awad, Rogoza [33] and Christodoulou, Markopoulou [34] used

**Table 1. Characteristics of included studies.**

| Author, year and country | Main objective | Study design | Sample characteristics | Assessment tools | Overall quality rating |
|---|---|---|---|---|---|
| Awad et al., 2022 [33]; Lebanon | To examine the relationship between SMUD and ON, and physical activity, as well as the indirect role of loneliness | Quantitative, cross-sectional | Sample: 363 university students<br>Mean age: 22.65 years<br>Gender: all; 61.7% women | -TOS<br>-SMD | 6 |
| Asil et al., 2023 [23]; Türkiye | To examine the relationship between orthorexic tendencies, social media users and the factors affecting these tendencies | Quantitative, cross-sectional | Sample: 2526 internet users<br>Age range: 18–65 years<br>Mean age: 28.4 years<br>Gender: all; 27.6% men | -ORTO-11<br>-SMEB | 7 |
| Cheshire et al., 2020 [22]; USA and UK | To understand the key defining ON features and what factors might influence its development | Qualitative, cross-sectional | Sample: 9 professionals working with those with ON tendencies (dieticians, clinical psychologists and a family therapist)<br>Mean age: 36.7 years<br>Gender: women | -Interviews | 7 |
| Christodoulou et al., 2024 [34]; Greece | To explore pathways through which mindful eating practices and Instagram use may influence the development and perpetuation of ON | Quantitative, cross-sectional | Sample: 407 Instagram users<br>Age range: 18–65 years<br>Gender: all; 68.3% women, 31% men and 0.7% non-binary | -ORTO-R<br>-Self-reported data about Instagram use | 5 |
| Geise, 2021 [35]; Germany, The Netherlands and Other | To examine the relationship between ON and social media, particularly focusing on healthy eating vlogs | Quantitative, cross-sectional | Sample: 236 university students<br>Mean age: 21.57 years<br>Gender: all; 83.2% women, 16.8% men | -ORTO-15<br>-Self-reported data about YouTube consumption; SMMS | 6 |
| Gabriel, 2021 [36]; Germany, the Netherlands, Italy, Romania and Finland | To examine the relationship between self-reported levels of PA, EA and ON-related symptoms in university students | Quantitative, cross-sectional | Sample: 232 university students<br>Age range: 17–38 years<br>Gender: all; 83.2% women, 16.8% men | -ORTO-15<br>-Self-reported data about social media use | 6 |
| Gobin et al., 2021 [24]; Canada | To examine the relationship between reported changes to adults' eating habits, exercise, and social media use during the COVID-19 pandemic and a lockdown period | Quantitative, cross-sectional | Sample: 143 participants<br>Age range: 17–73 years<br>Gender: women | -EHQ<br>-Self-reported data about social media use | 5 |
| Greene et al., 2023 [37] | To compare the presentation of eating disorders and pro-recovery communities related to eating disorders on TikTok | Mixed-methods, cross-sectional | Sample: 241 videos from users of TikTok who are involved in pro-recovery communities related to eating disorders | Content analysis of TikTok | 7 |
| Issa, 2022 [25]; Lebanon | To examine the relationship between social media use and dependence on dietary intake and prevalence of ON in pregnant women, and compare it to non-pregnant Lebanese women | Mixed-methods, case control | Sample: 490 women (250 were pregnant and 240 were not pregnant)<br>Age range: 19–46 years<br>Gender: women | -ORTO-15<br>-Subscales from MTUAS, including social media usage, online friendships, and social media anxiety/dependence | 6 |

*(Continued)*

**Table 1.** (Continued)

| Author, year and country | Main objective | Study design | Sample characteristics | Assessment tools | Overall quality rating |
|---|---|---|---|---|---|
| Kardaş, 2021 [38]; Türkiye | To examine the relationship between social media use, body perception and possible Orthorexia | Quantitative, cross-sectional | Sample: 148 students of Istanbul Gelisim University Age range: 18–37 years Gender: all; 50% women | -ORTO-11 -BSMAS | 6 |
| Karniej et al., 2023 [30]; Poland and Spain | To identify demographic factors associated with ON behavior among gay men | Quantitative, cross-sectional | Sample: 394 gay men Age range: 18–67 years Gender: men | -ORTO-15 -Self-reported data about social media use and Grindr dating app use | 7 |
| Levin et al., 2023 [39]; Canada | To examine the relationship between ON and clinically and theoretically related forms of psychopathology to assess whether ON should be classified as an ED, a variant of OCD, a symptom of OCPD, or a subtype of health anxiety | Quantitative, cross-sectional | Sample: 333 university students taking an Introduction to Psychology course Age range: 17.3–47.3 years Gender: all; 72% women | -EHQ -Self-reported data about social media use | 5 |
| Norton, 2018 [26]; Australia | To examine the relationship between Instagram use, personality factors and Orthorexia-like behavior in women | Quantitative, cross-sectional | Sample: 206 currently using Instagram and had never received an eating disorder diagnosis Age range: 18–57 years Gender: women | -ORTO-15 -MC-SDS; Facebook Social Connectedness Scale; Instagram Investment Inventory | 5 |
| Optiz et al., 2022 [40] | To explore individual experiences and conceptualizations of ON regarding characteristics and underlying causes, as outlined in ED- and diet-related subreddit communities, considering the online context in which comments were posted | Qualitative, cross-sectional | Sample: 7 subreddits that include Reddit users discussing ON | Content analysis of Reddit | 7 |
| Özkefeli Hamurcu et al., 2023 [41]; Türkiye | To examine the relationship between nursing students' social media use and their ON tendency | Quantitative. cross-sectional | Sample: 1191 undergraduate nursing students Age range: 18–37 years Mean age: 20.74 Gender: all; 84.4% women | -ORTO-11 -SMUIS | 7 |
| Piko et al., 2024 [27]; Hungaria | To examine orthorexic tendency in a sample of Hungarian female young adults with special interest in a healthy lifestyle and assess background variables | Quantitative, cross-sectional | Sample: 310 young adult women with a special interest in a healthy lifestyle Age range: 18–35 years Gender: women | -ORTO-15-BSMAS (Hungarian validated 6-item version) | 6 |
| Ross Arguedas, 2022 [42]; USA, Canada, UK, Estonia, Mexico and the Netherlands | To examine how health and medical knowledge, particularly about ON, is subverted and reshaped in the Orthorexia recovery community on Instagram | Qualitative, cross-sectional | Sample: 34 Instagram users who are part of the Orthorexia recovery community (individuals in recovery, recovered individuals, coaches, and some health professionals) Age range: 18–47 years Gender: all | Interviews | 6 |

*(Continued)*

**Table 1.** (Continued)

| Author, year and country | Main objective | Study design | Sample characteristics | Assessment tools | Overall quality rating |
|---|---|---|---|---|---|
| Santarossa et al., 2019 [43] | To investigate the #Orthorexia conversation on Instagram | Mixed-methods, cross-sectional | Sample: 145 unique images | -Content analysis of Instagram -Author biography analysis | 4 |
| Scheiber et al., 2023 [44]; Germany and Austria | To develop and empirically test a socio-cultural model for ON, which explicitly focuses on the influence of social media and whether social media directly influences ON | Mixed-methods, cross-sectional | Sample: 647 from the general population and participants with higher ON tendencies Age range: 18–30 years Gender: all; 54.6% women, 44.8% men and 0.6% diverse | -DOS (shortened 6-item version) -Measured using three items based on Schaefer, Terlutter, and Diehl (2019) and adapted to the SM context | 7 |
| Sener et al., 2023 [45]; Türkiye | To examine the relationship between social media use and/or addiction and the incidence of ON in individuals with a BMI of 25 or higher | Quantitative, cross-sectional | Sample: 147 individuals who applied to the Polyclinic (Başakşehir Çam Sakura City Hospital Family Medicine Clinic) and have a BMI value of 25 or more Age range: 18–65 years Gender: all; 67.2% women, 32.8% men | -ORTO-11 -SMBÖ-YF | 5 |
| Silva et al., 2023 [46]; Portugal and Brazil | To examine the relationship between social and personal factors and body dysmorphia and ON | Quantitative, cross-sectional | Sample: 498 Instagram users Age range: 18–50 years Gender: all; 64% women | -ORTO-15 -Self-reported data about Instagram use | 5 |
| Tarsitano et al., 2022 [47]; Italy | To examine the relationship between social media network use and ON | Quantitative, cross-sectional | Sample: 4111 Instagram user Mean age: 31 years Gender: all; 95.2% women, 3.9% men and 0.9% undeclared | -I-DOS -Self-reported data about Instagram use | 6 |
| Turner et al., 2017 [12]; UK, USA and Other | To examine the relationship between Instagram use and symptoms of ON | Quantitative, cross-sectional | Sample: 680 Instagram user Age range: 18–75 years Gender: women | -ORTO-15 -Self-reported data about social media use | 6 |
| Valente et al., 2020 [48] | To identify the biological, psychological, and social factors that contribute to the progression and developmental pathway of ON | Mixed-methods, cross-sectional | Sample: 185 people who self-diagnose, and others who do not self-diagnose but still post about ON Age range: 16–55 years Gender: all; 95% women (from the quantitative sample) | -Self-diagnosis through Instagram hashtags (#orthorexia) -Content analysis of Instagram posts about ON | 7 |
| Villa et al., 2022 [49]; Chile | To examine the associated risk factors for ON in Nutrition students | Quantitative, cross-sectional | Sample: 90 nutrition students Mean age: 22.2 years Gender: all; 87.8% women, 12.2% men | -ORTO-11-ES -Data about duration of Instagram use was obtained from participants' app settings | 6 |

*(Continued)*

**Table 1.** (Continued)

| Author, year and country | Main objective | Study design | Sample characteristics | Assessment tools | Overall quality rating |
|---|---|---|---|---|---|
| Yenidunya, 2021 [50]; Türkiye | To examine the relationship between socio-demographic variables, interpersonal attachment styles, social media addiction level and ON levels | Quantitative, cross-sectional | Sample: 384 active social media users over the age of 18 Age: over 18 years Gender: all; 79% women | -ORTO-11 (Turkish version) -SMBÖ-YF | 5 |
| Yılmaz et al., 2024 [51]; Türkiye | To examine the relationship between orthorexic symptoms in individuals and social media addiction, body image and eating attitude in the general population | Quantitative, cross-sectional | Sample: 345 participants from the general population, excluding those with a mental illness Age range: 18–65 years Gender: all; 47.2% women, 52.8% men | -ONI -SMBÖ | 5 |
| Yilmazel, 2021 [29]; Türkiye | To examine the relationship between Orthorexia tendency and social media addiction among candidate doctor and nurse populations | Quantitative, cross-sectional | Sample: 969 medical and nursing students of a public university Mean age: 21.4 year Gender: all; 53.8% men | -ORTO-15 -SMAS | 6 |
| Yilmazel et al., 2020 [52]; Türkiye | To examine the relationship between health literacy and ON among urban schoolteachers | Quantitative, cross-sectional | Sample: 442 school teachers Mean age: 43.4 years Gender: all; 63.9% women | -ORTO-15 -Self-reported data about Instagram use; TSOY-32 | 7 |
| Yurtdas-Depboylu et al., 2022 [53]; Türkiye | To examine the relationship between social media addiction and body image, ON, and eating attitude to determine risk factors associated with eating behavior disorder risk and ON tendency | Quantitative, cross-sectional | Sample: 1232 social media users Age range: 13–18 years Gender: all; 57.8% girls, 42.2% boys | -ORTO-11 -SMASA | 5 |
| Zemlyanskaya et al., 2022 [54]; Russia, Türkiye and Ukraine | To explore the conversation around ON on Instagram from the perspective of Russian speaking users | Mixed-methods, cross-sectional | Sample: 95 Russian-speaking Instagram users who post about ON Mean age: 29 years Gender: all; 94.4% women, 5.6% men | -Questionnaire created by the author | 7 |

regression analyses to investigate the relationship between the duration of social media use and ON. In addition to these, Opitz, Newman [40] and Greene, Norling [37] used qualitative and mixed methods approaches to explore the cultural and psychological underpinnings of ON in online communities. Mostly, quantitative studies used the ORTO-11, [55] ORTO-15, [31] and Düsseldorf Orthorexia Scale (DOS) [56] to confirm ON tendencies while qualitative studies were based on thematic analyses on the social media content or interview transcripts. In studies that examined social media behaviors, the metrics that were studied assessed the duration of screen time, frequency of platform visits, and engagement with health related content. A few studies used a mixed methods approach to collect quantitative and qualitative data. Greene, Norling [37] employed a content analysis combined with surveys in evaluating the impact of TikTok pro-recovery communities in their shaping of users' eating disorders attitudes. The qualitative and mixed-methods studies included in the current review are methodologically solid with minor limitations. Meanwhile, methodological limitations in the included quantitative studies suggest moderate quality. The quality of the included studies is adequate (Table 2).

**Table 2. Methodological quality assessment for included studies.**

**Methodological quality assessment for included quantitative studies**

| Author, year | Are there clear research questions? | Do the collected data allow to address the research questions? | Is the qualitative approach appropriate to answer the research question? | Are the qualitative data collection methods adequate to address the research question? | Are the findings adequately derived from the data? | Is the interpretation of results sufficiently substantiated by data? | Is there coherence between qualitative data sources, collection, analysis and interpretation? |
|---|---|---|---|---|---|---|---|
| Cheshire et al., 2020 [22] | Yes | Yes | Yes | Yes | Yes | Yes | Yes |
| Optiz et al., 2022 [40] | Yes | Yes | Yes | Yes | Yes | Yes | Yes |
| Ross Arguedas, 2022 [42] | No | Yes | Yes | Yes | Yes | Yes | Yes |

**Methodological Quality Assessment for Included Quantitative Studies**

| Author, year | Are there clear research questions? | Do the collected data allow to address the research questions? | Are the participants representative of the target population? | Are measurements appropriate regarding both the outcome and intervention (or exposure)? | Are there complete outcome data? | Are possible confounders accounted for in the design and analysis? | During the study period, is the intervention administered (or exposure occurred) as intended? |
|---|---|---|---|---|---|---|---|
| Awad et al., 2022 [33] | Yes | Yes | Yes | No | Yes | Yes | Yes |
| Asil et al., 2023 [23] | Yes | Yes | Yes | Yes | Yes | Yes | Yes |
| Christodoulou et al., 2024 [34] | Yes | Yes | Yes | No | Yes | No | Yes |
| Geise, 2021 [35] | Yes | Yes | Yes | No | Yes | Yes | Yes |
| Gabriel, 2021 [36] | Yes | Yes | Yes | No | Yes | Yes | Yes |
| Gobin et al., 2021 [24] | Yes | Yes | Yes | No | Yes | No | Yes |
| Kardaş, 2021 [38] | Yes | Yes | Yes | No | Yes | Yes | Yes |
| Karniej et al., 2023 [30] | Yes | Yes | Yes | Yes | Yes | Yes | Yes |
| Levin et al., 2023 [39] | Yes | Yes | No | No | Yes | Yes | Yes |
| Norton, 2018 [26] | Yes | Yes | Yes | No | Yes | No | Yes |
| Özkefeli Hamurcu et al., 2023 [41] | Yes | Yes | Yes | Yes | Yes | Yes | Yes |
| Piko et al., 2024 [27] | Yes | Yes | Yes | Yes | Yes | No | Yes |
| Sener et al., 2023 [45] | Yes | Yes | No | Yes | Yes | No | Yes |
| Silva et al., 2023 [46] | Yes | Yes | No | Yes | Yes | No | Yes |

*(Continued)*

**Table 2.** (Continued)

**Methodological quality assessment for included quantitative studies**

| Author, year | Are there clear research questions? | Do the collected data allow to address the research questions? | Is the qualitative approach appropriate to answer the research question? | Are the qualitative data collection methods adequate to address the research question? | Are the findings adequately derived from the data? | Is the interpretation of results sufficiently substantiated by data? | Is there coherence between qualitative data sources, collection, analysis and interpretation? |
|---|---|---|---|---|---|---|---|
| Tarsitano et al., 2022 [47] | Yes | Yes | No | Yes | Yes | Yes | Yes |
| Turner et al., 2017 [12] | Yes | Yes | No | Yes | Yes | Yes | Yes |
| Villa et al., 2022 [49] | Yes | Yes | Yes | Yes | Yes | No | Yes |
| Yenidunya, 2021 [50] | Yes | Yes | No | Yes | Yes | No | Yes |
| Yılmaz et al., 2024 [51] | Yes | Yes | Yes | Yes | Can't tell | No | Yes |
| Yilmazel, 2021 [29] | Yes | Yes | No | Yes | Yes | Yes | Yes |
| Yilmazel et al., 2020 [52] | Yes | Yes | Yes | Yes | Yes | Yes | Yes |
| Yurtdas-Depboylu et al., 2022 [53] | Yes | Yes | Yes | Yes | Can't tell | No | Yes |

**Methodological Quality Assessment for Included Mixed-Methods Studies**

| Author, year | Are there clear research questions? | Do the collected data allow to address the research questions? | Is there an adequate rationale for using a mixed-methods design to address the research question? | Are the different components of the study effectively integrated to answer the research question? | Are the outputs of the integration of qualitative and quantitative components adequately interpreted? | Are divergences and inconsistencies between quantitative and qualitative results adequately addressed? | Do the different components of the study adhere to the quality criteria of each tradition of the methods involved? |
|---|---|---|---|---|---|---|---|
| Greene et al., 2023 [37] | Yes | Yes | Yes | Yes | Yes | Yes | Yes |
| Issa, 2022 [25] | Yes | Yes | Yes | Yes | Yes | Yes | Yes |
| Santarossa et al., 2019 [43] | Yes | Yes | No | Yes | Yes | No | Can't tell |
| Scheiber et al., 2023 [44] | Yes | Yes | Yes | Yes | Yes | Yes | Yes |
| Valente et al., 2020 [48] | Yes | Yes | Yes | Yes | Yes | Yes | Yes |
| Zemlyanskaya et al., 2022 [54] | Yes | Yes | Yes | Yes | Yes | Yes | Yes |

## Results of included studies

### Social media platforms

A frequent finding among included studies was that the use of specific social media platforms was associated with higher ON risk. ON was most commonly associated with Instagram use [12,26,33,49]- [57]. Awad, Rogoza [33] also discovered that Instagram use, specifically visually based content produced by fitness-focused content creators, was positively

correlated with ON levels. Awad, Rogoza [33] also identified a positive relationship between Tumblr use and ON, and that Tumblr users engaged in communities promoting wellness or dietary purity showed more ON symptoms. Additionally, exposure to Instagram, particularly 'fitspiration' content that presents idealized, health-centered lifestyles and appearance focused health goals, predicted ON in women [26]. Turner and Lefevre [12] also found a positive relationship between Instagram use and ON symptoms, but other platforms such as Facebook and Twitter had weaker or non-significant relationships. Meanwhile, Santarossa, Lacasse [43] found that Instagram use and exposure to the ON community of recovered individuals on the platform can provide support for recovery, highlighting the potentially positive influence of visually-immersive social networking such as Instagram. On YouTube, people who frequently watch health related vlogs or diet related content had more ON symptoms [35], possibly because of exposure to extreme dietary advice or visually appealing portrayals of healthy eating. This platform specificity shows how the social media structure and user experience can affect people differently in terms of ON, especially visually immersive and community-based platforms.

### Duration of social media use

The time spent on social media was another important factor related to ON. Previously, four studies have shown that people who spend a lot of time on social media have higher ON symptoms irrespective of the type of platform.[27,29,41,44]. In Tarsitano, Pujia [47] people who spent more than 60 minutes per day on social media had higher ON symptoms than those who used less than 15 minutes per day. Awad, Rogoza [33] found that people who spent between 30 and 60 minutes on social media per day had higher ON levels than those who used less than 30 minutes. Other studies showed that longer time on social media may increase the chances of exposure to ON promoting content such as dietary advice, fitness ideals and images of food perceived as healthy [53]. Hamurcu and Yılmaz [41] reported that ON levels are higher among nursing students who use social media more frequently, suggesting that more prolonged exposure, especially for potentially more health-conscious groups, might increase ON risk. Also, Yılmazel [29] discovered that those with higher social media addiction, as measured by prolonged usage and difficulty disengaging from social media, had more ON symptoms.

### Social media content

Health-related and fitness related content showed the strongest associations with increased ON risk.[23,26] Several studies have pointed out that ON risk is increased when exposed to nutrition focused posts, including 'clean eating' or dietary restrictions such as which foods to eat.[44,53] Asil, Yılmaz [23] found that individuals who consumed nutrition or fitness related content were more likely to report ON symptoms, which suggests that some themes on social media can encourage ON behavior. Furthermore, people who consumed health and fitness focused social media posts were more likely to have ON symptoms, according to Scheiber, Diehl [44] These same posts often revolved around physical perfection, self-discipline and strict dietary regimes, which can reflect ON traits as well.[44] Within the same realm, it can be hypothesized that individuals exhibiting ON symptoms might be more likely to consume social media content related to healthy eating and fitness ideals. Norton [26] showed that users that reported viewing content from fitness "micro-celebrities" on Instagram exhibited higher ON. In Greene, Norling [37] posts using the hashtag "#orthorexiarecovery" often included discussion about diet culture and perhaps provided support to people who were trying to get over ON. Taken together, these findings suggest that the thematic content of social media is associated with ON risk, of which diet and fitness themes are particularly influential.

### Individual-level factors

ON was related to social media, and important moderators in this relationship were demographic factors, including gender, education level and specific life stages. In Yilmazel and Bozdogan [52] users of Instagram who are women exhibiting lower health literacy reported higher ON symptoms indicating that vulnerabilities such as limited health knowledge may

increase ON risk for women. In Issa [25] ON was found to be more prevalent in pregnant women than in non-pregnant women, but social media use was not associated with ON in the former subgroup. Kardaş [38] indicated that higher social media use and higher ON symptoms are often displayed by young adults and adolescents as a result of the developmental stages associated with heightened social comparison and body image pressures. Silva, Elmashhara [46] found that for Brazilian and Portuguese people, body dissatisfaction mediated the association between Instagram use and ON, indicating that sociocultural perceptions of body image and health may influence the relationship between Instagram use and ON. These results suggest that individual-level variables may, at least in part, affect the relation between ON and social media use.

### Risk of bias assessment

Out of the reviewed studies, as detailed in Table 3, 15 reported a high risk of bias, driven primarily by significant methodological limitations. Most studies used cross sectional designs, which limits causal inferences from social media to ON. As a result, it is unknown whether social media exposure causes ON symptoms directly, or whether people with ON are more likely to engage with certain types of social media. A second limitation might be reliance on self-report measures that can generate response biases and inaccuracies in reporting ON symptoms and social media behaviors. Yet, it is most reasonable to ask individuals to self-report ON symptoms. In addition, many of the studies used non-representative samples, and were usually not representative of men nor specific cultural groups, thereby limiting the generalizability of the results to broader populations. Furthermore, there is no validated tool to measure ON across studies or social media addiction. ORTO-11 and ORTO-15 were frequently used, but have been criticized for low internal consistency and yielding different psychometric properties among varying cultures [23,35]. In some studies, new social media use scales were used, including self-constructed items Geise [35,36,57], which might require further research to assess validity and reliability.

## Discussion

The current scoping review on ON and social media identified 31 studies between 2017 and 2024. The methodology used varied including 22 quantitative, three qualitative, and six mixed-methods studies. All included studies adopted a cross-sectional design, except for one case control study. The included studies were conducted in different continents including Asia, Europe, North and South America, and Australia. It is noticeable that the investigation of ON and social media has been attracting increasing attention in the past few years. Social media platforms focused on imagery, such as Instagram, YouTube, and TikTok, were investigated of 11 included studies. Social media use duration was the focal point of nine of the included studies while 13 studies focused on the content. Our findings provide a synthesis of the relationship between ON and the use of social media in various aspects such as type of platform, duration of use and content. It also explores the interplay of ON, social media, and individual-level factors.

### ON and social media platforms

The use of platforms that offer predominantly visual content, such as photos and videos, including YouTube, Instagram and Tumblr, showed strong associations with ON, in comparison with the use of social media networks that mainly offer text content, such as Facebook and Twitter [12,33–35,49,52]. Previous research has indeed shown that image-focused social media platforms were more likely to influence social media users, irrespective of whether it promoted positive or maladaptive eating behaviors [57]. Specifically, content on Instagram showing "fitspiration" emphasized how fitness-related content can relate to problematic eating behaviors such as ON [23,26,28,36,44] Social media platforms work according to specific and highly complex algorithms, which will heavily promote different types of content that the user seems to be interested in, generating tailored content recommendations with the end goal of maximizing the time individuals spend on the platform [58]. Through these mechanisms, the consistent exposure to ON-reinforcing content could potentially worsen the risks of adopting maladaptive eating behaviors for users who are interested.

**Table 3. Results of included studies.**

| Direction of relationship | Author, year | Main results | Limitations | Conclusion |
|---|---|---|---|---|
| ON→SM | Greene et al., 2023 [37] | -Similar discussion was found between the posts with the hashtags #anarecovery, #arfdrecovery, #bedrecovery, #miarecovery, and #orthorexiarecovery including emphasis on recovery as a process.<br>-#orthorexiarecovery focused on critiquing diet culture. | -The study focused on only five discrete hashtags, potentially excluding a broader representation of the TikTok pro-recovery community.<br>-The study was restricted to only publicly available, English-language videos.<br>-The emphasis was on popular content, which may not reflect trends among creators with smaller followings.<br>-The study excluded minors, a significant user base on TikTok. | Eating disorders recovery portrayal on TikTok differs according to the diagnostic label. |
| | Levin et al., 2023 [39] | Individuals exhibiting ON and eating pathology, with or without weight or shape concerns, showed a higher probability of consuming ($p<.01$) and sharing healthy eating content on social media ($p<.001$). | -The results are not generalizable.<br>-ON may be related to OCPD factors, such as perfectionism and cognitive rigidity while only an overall OCPD score was used in this study. | ON shows an overlap with eating pathology, and consuming and sharing health-related content on social media is associated with ON and eating pathology. |
| | Ross Arguedas, 2022 [42] | Instagram involves interaction through imagery which gives its recovery community a quality. | -The sample is not representative.<br>-The study focused only on Instagram, and results may not generalizable to other platforms. | Community can validate the experiences of individuals that exhibit ON despite the lack of its medicalization currently. |
| | Santarossa et al., 2019 [43] | -Love ($n=535$) and #edrecovery ($n=425$) were the most used words from the collected records.<br>-Most authors of posts were women (84%) who discussed recovery from maladaptive eating behaviors. | -There is risk of bias due to the use of self-report measures.<br>-There is difficulty in differentiating between healthy eating and orthorexic behaviors.<br>-The study focused only on Instagram, and results may not generalizable to other platforms. | The study suggests that the ON community on Instagram is supportive of recovery. |
| | Valente et al., 2020 [48] | -Participants reported interacting factors regarding onset of ON.<br>-Obsession with healthy eating was the most reported symptom.<br>-Participants wanted to lose weight for health reasons. | -The sample is not representative.<br>-There is risk of bias due to the use of self-report measures.<br>-The cross-sectional design limits causality.<br>-The study focused only on Instagram, and results may not generalizable to other platforms. | The study provides personal perspectives of ON. |
| | Yilmaz et al., 2024 [51] | Participants consuming healthy eating programs on social media exhibited higher ON in comparison to those who were not ($p<0.05$). | -The cross-sectional design limits causality.<br>-There is risk of bias due to the use of self-report measures.<br>-The study may not account for the effects of different types of social media platforms, as it groups them under one category.<br>-The motivation for using social media was not questioned.<br>-The results are not generalizable.<br>-The Cronbach's alpha values of the scales used were not calculated. | Consuming content about healthy eating on social media is associated with higher risk of ON. |
| | Yilmazel, 2021 [29] | 31% of individuals who had high levels of social media addiction exhibited ON. | -The cross-sectional design limits causality.<br>-The sample is not representative. | Social media addiction is associated with higher ON. |
| | Zemlyanskaya et al., 2022 [54] | Instagram can have a positive or negative effect on ON. It can expose individuals to beauty ideals and diets, but can promote recovery in other communities. | -The results are not generalizable.<br>-There is risk of bias due to the use of self-report measures.<br>-The content analysis was limited to a specific time, which may not capture the full range of discussions and trends related to ON.<br>-The use of English hashtags by Russian-speakers makes it difficult to identify socio-cultural differences. | Instagram can be a triggering factor for ON or a support in recovery. |

*(Continued)*

| Direction of relationship | Author, year | Main results | Limitations | Conclusion |
|---|---|---|---|---|
| SM→ON | Christodoulou et al., 2024 [34] | Extensive use of Instagram had a positive relationship with ON (p<0.05). | -The cross-sectional design limits causality.<br>-There is risk of bias due to the use of self-report measures.<br>-The sample is not representative. | Mindful eating could be a protective factor against ON, and Instagram use could be a risk factor for ON. |
| | Kardaş, 2021 [38] | Orthorexia and social media have a weak but significant relationship (r=−0.116, p<0.05). | -The sample is not representative.<br>-It is not known exactly how much time is spent on social media.<br>-There is risk of bias due to the use of self-report measures. | Social media use was significantly associated with higher ON, but the association is weak in this study. |
| | Karniej et al., 2023 [30] | -The use of the Grindr application (OR=5.312, 95%CI, 3.373–8.365) predicted ON.<br>-Instagram users had lower risk of ON (OR=0.479, 95%CI, 0.279–0.822). | -The sample is not representative.<br>-The cross-sectional design limits causality.<br>-It's difficult to define unequivocally whether a person with a high score in the ORTO represents only a specific life-style characterizing in a healthy eating habits or reflects pathological eating.<br>-Researchers did not have data from study participants related to auto immune diagnosis, gastrointestinal problems, health related anxiety, and other impacting psychological constructs that can be related to ON behavior. | Using Grindr dating app is associated with higher ON among gay men, while Instagram was not. |
| | Norton, 2018 [26] | -Benign envy, appearance comparison, and exposure to micro-celebrities on social media predicted 42.2% of orthorexia behavior among women.<br>-Individuals who viewed health-related content exhibited higher ON (p=.012).<br>-Women who were exposed to micro-celebrities on Instagram exhibited higher ON (p<.001).<br>-Women who were exposed to micro-celebrities posting fitness and healthy food content on social media exhibited higher ON (M=16.50, SD=4.77). | -A general definition of fitspiration used, making it unclear if posts included weight loss promotion, thin-ideal body imagery, or appearance-focused health content.<br>-A general measurement of Instagram connectedness was used, not accounting for specific types of communities or participation behaviors (posting, viewing, commenting).<br>-There is risk of bias due to the use of self-report measures.<br>-The cross-sectional design limits causality.<br>-The focus on male body image is limited. | Exposure to certain content on Instagram such as micro-celebrities is associated with a higher risk of ON among women. |
| | Opitz et al., 2022 [40] | -Seeing food as a threat and pursuing superiority were associated with ON.<br>-Some commenters viewed ON as a safety mechanism and selfcare.<br>-Some topics discussed on diet-related subreddits were ON as health perfectionism and ON as a pathology. | -The study focused solely on comments explicitly mentioning "orthorexia," excluding those describing ON without using the term.<br>-The study analyzed only comments and not original posts or comment authors.<br>-The study did not include discussions from the subreddit r/Orthorexia, which was underused at the time. | ON conceptualization must account for individual and sociocultural factors. |
| | Özkefeli Hamurcu et al., 2023 [41] | There was no difference between ON levels across different social media platforms, but a difference was observed according to the duration of social media use (r=−0.11; p=0.04). | -There is risk of bias due to the use of self-report measures.<br>-The sample is not representative. | An increased duration of social media use is associated with higher ON among nursing students. |
| | Piko et al., 2024 [27] | Eating disorder attitudes, physical activity, social media addiction and drive for thinness (β=0.54, p<0.001) were significantly associated with ON. | -The cross-sectional design limits causality.<br>-There is risk of bias due to the use of self-report measures.<br>-The sample is not representative. | Social media addiction is associated with higher ON among Hungarian women. |

**Table 3.** (Continued)

| Direction of relationship | Author, year | Main results | Limitations | Conclusion |
|---|---|---|---|---|
| | Scheiber et al., 2023 [44] | Social media users who consume health and fitness content exhibit higher ON ($\chi$ = 0.204, $p$ < 0.001). | -The cross-sectional design limits causality.<br>-There is risk of bias due to the use of self-report measures.<br>-The sample is not representative.<br>-The study focused only on Instagram, and results may not generalizable to other platforms. | Interacting with specific social media content such as health and fitness is associated with higher ON. |
| | Sener et al., 2023 [45] | Individuals who scored higher on the virtual communication aspect of social media addiction exhibited more ON ($rho$=-0.187; $p$ = .013). | -The results are not generalizable.<br>-The sample is not representative.<br>-The cross-sectional design limits causality. | Virtual communication through social media is associated with higher ON levels. |
| | Silva et al., 2023 [46] | -Body dissatisfaction mediated the relationship between Instagram use and ON ($p$ = 0.000).<br>-No cultural differences were found between the Brazilian and Portuguese samples. | -The sample is not representative.<br>-The study focused only on Instagram, and results may not generalizable to other platforms.<br>-The cross-sectional design limits causality.<br>-There is risk of bias due to the use of self-report measures. | Body dissatisfaction plays a role in the association between Instagram use and ON. |
| | Tarsitano et al., 2022 [47] | Participants who reported using social media for over 60 minutes per day had a higher prevalence of ON than those using social media for less than 15 minutes per day ($p$ < 0.001). | -The sample is not representative.<br>-There is risk of bias due to the use of self-report measures. | A longer duration of social media use is associated with higher ON. |
| | Turner et al., 2017 [12] | Higher Instagram use was associated with a greater tendency towards ON ($r$ = -0.10; $p$ = 0.01), while other platforms were not. | -The sample is not representative.<br>-There is risk of bias due to the use of self-report measures.<br>-The cross-sectional design limits causality.<br>-The study did not account for potential interactions between the multiple social media platforms used by the participants.<br>-The study did not assess whether participants followed a strict diet for medical reasons. | Instagram content has a more significant relationship with ON in comparison with other social media platforms. |
| | Villa et al., 2022 [49] | Limited physical activity (Sedentary OR 2.42, Heavy OR 3.53) and time spent on the social network Instagram (<1h OR 2.77, >3h OR 1.80) were associated with ON risk. | -The results are not generalizable.<br>-The cross-sectional design limits causality.<br>-There was a lack of comparison with other populations or regions.<br>-The ORTO-11-ES questionnaire used has different versions, which complicates comparisons with other studies using different tools. | Physical activity and Instagram use could be risk factors for ON among nutrition students. |
| | Yenidunya, 202 [50] | Social media addiction was associated with higher ON levels ($t$ = -2,384). | -The cross-sectional design limits causality.<br>-The sample is not representative.<br>-There is risk of bias due to the use of self-report measures.<br>-The study may not account for the effects of different types of social media platforms, as it groups them under one category. | A positive relationship was found between social media addiction and ON. |
| | Yilmazel et al., 2020 [52] | Being a woman (aOR=0.51;95% C.I. = 0.29–0.91; p = 0.041), using Instagram (aOR=1.69;95% C.I. = 1.04–2.75; p = 0.033) and having limited health literacy (aOR=4.85;95% C.I. = 2.15–10.94; p = 0.001) was associated with ON. | -It is difficult to determine the TSOY-32's efficacy in the absence of a global health literacy scale.<br>-The cross-sectional design limits causality.<br>-The sample is not representative. | Users of Instagram who are women and have limited health literacy might be at a higher risk of ON. |

*(Continued)*

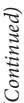

| Direction of relationship | Author, year | Main results | Limitations | Conclusion |
|---|---|---|---|---|
| | Yurtdas-Depboylu et al., 2022 [53] | -High social media addiction levels were associated with higher ON (OR: 1.02, 95% CI = 1.00–1.04).<br>-Adolescents who often consumed nutrition-related content on social media exhibited higher ON (OR: 5.49, 95% CI 3.39–8.88). | -The cross-sectional design limits causality.<br>-The results are not generalizable.<br>-There is risk of bias due to the use of self-report measures. | Exposure to nutrition content on social media and social media addiction are associated with higher risk of ON. |

ON: Orthorexia Nervosa; SM: Social Media; SMUD: Social Media Use Disorder

It is crucial to keep in mind that content published about health that seems harmless or beneficial could also contribute to ON. Many individuals on social media provide advice on how to lead a healthy lifestyle, nutritional guidance and work-out plans. Unfortunately, this content can backfire as it might promote adherence to these recommendations as a necessity for wellbeing. Users may accept these guidelines without realizing that they could develop into obsessive behaviors, which can unintentionally promote strict dietary guidelines and perfectionistic impulses. In these situations, the normalization of "healthy eating" can cause people to adopt more rigid and strict eating habits, making it harder to distinguish between ON and healthy habits. Moreover, carefully curated algorithms tend to reinforce these initially benign narratives and create echo chambers where only a limited notion of health is evident. This repeated exposure to health and fitness related content might intensify the perceived pressure to meet these expectations, turning seemingly innocent lifestyle messages into risk factors for the onset or worsening of ON.

## ON and duration of social media use

The duration of social media use was consistently associated with ON in the included studies: the longer time users spend on social media platforms, the higher the probability of elevated ON levels [29,33,41,47,50,53]. Potentially, these results could be due to the extended time that users spend browsing dietary content promoting purity and restriction on social media. Furthermore, the longer the duration of social media use, the more invested users may become in observing and participating in health-related content and communities, which could subsequently increase the risk of engaging in maladaptive eating behaviors [39]. Meanwhile, it is equally possible that individuals who have a higher susceptibility to ON are more likely to search for and engage with content that is focused on diet and health [27,39]. According to the uses and gratifications theory, individuals can seek information that meets their needs and interests through media [59]. As a result, social media users might spend an extended duration of time interacting with content that interests them on different platforms, further reinforcing social networking algorithms to offer relevant content [58]. Additionally, the current findings can be explained by confirmation bias, the tendency to prefer or accept information that conforms with our initial beliefs [60]. As a result, research has shown that confirmation bias played a role in both the information that we consume and share on social networks [60,61]. Overall, the relationship between ON and duration spent on social media is likely to be bidirectional, where these variables might reinforce each other.

## ON and social media content

The thematic content of social media posts, especially those related to health, fitness, and dietary purity, seems to be one of the most prominent factors associated with ON. Posts that promote "clean eating," self-discipline, and thinness tend to increase ON risks, whereas recovery-oriented content, such as "#orthorexiarecovery" hashtags, has mixed effects [37,40,42,57]. For example, many individuals who were exposed to ON recovery communities on social media were not aware of their maladaptive eating patterns [57], which might have led them to related these eating behaviors to symptoms of ON. Some social media users who came across ON recovery communities were able to self-diagnose and discover the source of their distress, however, this led them to identify as marginalized and ill [42]. Additionally, some recovery communities focused on criticizing diet culture [37], potentially exerting further audit on dietary behaviors that is in accordance with ON. These findings are in line with the literature on the harmful role of perfectionistic and restrictive dietary ideals in fostering disordered eating behaviors [62]. However, the algorithmic nature of social media may limit recovery-focused content's reach, unless individuals make purposeful goal-directed actions to adapt the algorithm over time by not engaging with fitness-related content.

That said, social media can be useful in the context of ON. Communities on social media that include people who have recovered from maladaptive eating behaviors such as ON might offer helpful psychological support for individuals who are looking to recover themselves. They might provide advice and strategy on how to adopt a balanced lifestyle through their own experience. Additionally, observing others who have identified and overcome patterns of maladaptive eating

can provide a sense of belongingness and encouragement. The growth of such recovery-oriented groups can encourage individuals with ON to think critically about the distinction between sustainable and balanced eating behaviors as opposed to unrealistic harmful ones. Additionally, social media platforms can assist users in redefining their connection with food and body image, potentially reducing the chance of developing ON, by giving attention to content that questions excessive dietary ideals and disseminates evidence-based nutritional recommendations. Therefore, these platforms have a special chance to empower users through community support and recovery-oriented information, even though the algorithmic amplification of unrealistic health and fitness content remains a concern.

### ON and individual-level factors

Moderating factors between ON and social media in the reviewed studies included gender, life stage, and cultural context. Individuals who reported being concerned for their health, exhibited obsessive and perfectionistic traits and were exposed to moral rigidity during their upbringing were more likely to report high ON tendencies [22,48]. Doing moderate and rigorous physical activity was also related to higher ON [36]. With respect to gender, women who used social media networks showed higher levels of ON [24,26,52]. indicating that appearance-related pressure experienced by women extends to diverse social domains, including social media [63]. As for age, the results suggest that ON levels were higher among adolescents and young adults in comparison with older adults, highlighting a common theme of potential appearance and lifestyle comparison within these age groups [30,53].

### Research gaps

Through the current scoping review, we were able to identify the existing research gaps in the literature about ON and social media. The included studies were observational and cross-sectional, indicating the scarcity of experimental studies and limiting the ability to synthesize findings identifying a cause-effect relationship between ON and social media. As previously presented, social media platforms can potentially have a mitigating or distressing effect on ON [57]. Experimental studies can help pinpoint the role of potential moderators such as social media use duration, content and platforms on ON symptomatology. Also, experimental designs where exposure to healthy food-related content on social media, as opposed to other unrelated content, can help determine potential causality. Adding to that, longitudinal studies can be conducted to assess the temporal frame of the relationship between ON and social media, tracking the progression or changes in orthorexic tendencies in parallel with social media use. Also, it can help identify different trajectories and profiles of participants and their characteristics through longitudinal designs. It is also important to note that only 3 qualitative studies about ON and social media were identified in the current review. Therefore, it is useful to conduct more qualitative research in order to gain insight into the reasons why individuals might look for content that promotes ON. Qualitative studies can investigate the narratives of people on the perceived effect of health and fitness content on their eating behaviors. Furthermore, it can relay the experience of individuals who have managed to overcome ON and the potential role of social media on their recovery.

Moreover, more than half of the included studies were conducted on sample from Westernized populations including the US and Canada, Western-Europe and Australia. Meanwhile, a smaller number of studies were conducted in Asian countries such as Türkiye, Russia (North Asia), and Lebanon. Only two included studies were done in South America, which involved participants from Brazil, Portugal, and Chile. Predominantly, the studies included in the review were in the English language. Subsequently, a major research gap exists in the investigation of the association between ON and social media among non-Westernized cultures. ON has been studied in many cultures, evident by the development and validation of different ON scales in developing countries such as the LONI in Arabic [64], DOS in Portuguese for the Brazilian population [65] and DOS for the Turkish population [51]. The current review highlights a scarcity in the study of ON and social media in different areas of the world such as the Middle East and North Africa region (MENA) that encompass developing and non-English speaking countries. A scoping review including 12 Arab countries from the MENA found that

31.4% of participants from included studies exhibited disordered eating attitudes [66]. Furthermore, a recent systematic review showed that the use of social media might be particularly problematic for mental health in the MENA [67]. These syntheses of the literature underline the need to conduct studies investigating ON and social media in the MENA, given the prevalence of maladaptive eating attitudes and the positive relationship between problematic social media and poorer mental health. Further research can help identify the effect that social media might have on ON specifically among individuals in the MENA.

It is also important to note that 21 of the included studies involved samples of adults only, meaning individuals over the age of 18, with only one study focusing on the relationship between ON and social media exclusively among adolescents aged between 13 and 18 [53]. The findings at hand portray a lack of studies evaluating ON symptomatology among adolescents and children in the context of social media use. A previous study showed that Polish adolescents illustrated a prevalence of ON similar to that of adults [68], implying that ON is a problematic eating behavior that is relevant to younger age groups. As our included studies mostly showed, social media use has a positive relationship with ON. Therefore, it is critical to conduct investigations on ON and social media use in adolescent samples, as social media use is higher for adolescents in comparison with young adults [69].

In addition, most included studies had either a majority of women participants or exclusively women participants, with only two studies having a slightly greater percentage of men within the sample [28,29] These results might indicate that women are more willing to participate in studies investigating ON in comparison with men [70]. Gabriel [36] found that men had more severe ON symptomatology in comparison to women, despite the sample having only 16.8% men. In general, women have a higher probability of reporting ON-related behavior than men [71]. Nevertheless, more studies need to assess ON and social media with larger samples of men as findings remain limited for them.

Another gap in available research is ambiguity about possible developmental pathways between ON and social media. In our scoping review, only two studies investigated mediating factors between ON and social media. Awad, Rogoza [33] found that loneliness only partially mediated the relationship between ON and Social Media Use Disorder (SMUD), while Silva, Elmashhara [46] showed that body dissatisfaction was a mediator between ON and Instagram use. Many factors that contribute to ON have been identified such as level of education, depressive symptoms [15], and adhering to vegan and vegetarian diets [72]. These factors might influence the direction and strength of the relationship involving ON and social media. Therefore, further research is needed to pinpoint potential contributing factors to the association between ON and social media. Specifically, experimental studies are required to identify factors in the causal pathway between ON and social media, further probing into a potential bidirectional relationship.

## Implications

The findings of the current scoping review have implications extending to different fields. First, the effects of social media content, platforms, and duration on ON levels might guide public and clinical interventions that could promote health and conduct ON preventative measures. Clinically, findings point to the relevance of social media use patterns in making diagnoses and setting treatment plans for ON. Diagnostic tools developed to screen for ON might ask about patterns of social media use, such as daily use and types of content exposure, considering its high relevance in today's age. For example, the LONI, a scale that was recently developed includes items such as "I follow influencers online because they help me come up with my own diet strategy." and "I have an online presence on social media because I want to help other people by influencing them with what I eat and believe" [64]. Lastly, social media literacy training may be an important component of treatment to assist individuals in critically analyzing information obtained online. For instance, the Butterfly Foundation from Australia has described several social media literacy programs that aid in mitigating the negative effects of social media use on eating behaviors and body image [73]. Researchers from the foundation advocate for safer social media regulations and make recommendations to the concerning authorities [73]. Additionally, they collaborate with different social media platforms on observing and reporting damaging content [73]. Algorithmic changes for social media that

promote more diverse and recovery-oriented content over extremist health and diet movements can make social media use beneficial instead of harmful. Governmental regulations are required to ensure that such changes are implemented and social media companies are kept accountable. For example, the European Union initiated the Digital Services Act (DSA) in order to monitor and hopefully diminish/discourage detrimental influences and activity online [74]. Content moderation policies in other areas in the world might seek to reduce the spread of destructive dietary ideals, while amplifying educational resources and community support for those vulnerable to ON.

## Strengths and limitations

The systematic methodology used in the current scoping review allows an extensive synthesis of the existing literature on ON and social media so far. The current scoping review followed the Preferred Reporting Items for Systematic reviews and Meta-Analyses (PRISMA) statement [17] and the PRISMA-S, which is an extension for literature search [18]. The PRISMA-ScR, which is an extension for scoping reviews was also referred to [19]. We used a sensitive search strategy to increase the probability of finding specific and precise research items. The present review was registered *a priori* in the OSF registries. The included studies were diverse in terms of location in the world, cultural groups making up the samples of participants, and language of the published research. Yet, a large number of included studies were conducted on Westernized populations, which shows that more research conducted among different cultural and geographical regions is needed and can help to provide a more comprehensive and holistic picture of the relationships between ON and social media.

Additionally, a comprehensive review was conducted of elements related to social media, such as content, platforms and duration, instead of exploring social media in general, making the findings more specific. Prior to the conclusion of the study, a literature review was performed to ensure that all relevant studies were evaluated. All studies were assessed for quality using the MMAT Version 18 [20].

First, 19 the included studies adopted a convenience sampling method without conducting a sample size calculation, constricting the generalizability of the results within the specified population. Subsequently, it reduces the ability to derive generally applicable conclusion from the scoping review. More than half of the included studies did not report a response rate, nor was information about participants who did not respond mentioned.

The scales used in the included studies differed, while most used the ORTO-11 and ORTO-15. However, the cut-off point of these two scales was inconsistent across studies [25,30,36] Having said that, the psychometric quality of different ORTO versions, especially the first version being ORTO-15, were characterized by low internal consistency [75,76]. In sum, because studies use different scales to measure orthorexic tendencies and the suboptimal psychometric qualities of some of these scales, the possibility to compare the outcomes across studies is limited. The development and validation of new scales to measure orthorexic tendencies could ensure higher reliability of assessment.

A similar limitation is that included studies used diverse scales to assess social media use. While some studies included self-reported measures created by the author to measure social media use, others used different scales [27,28,50] Therefore, the evidence derived from the included studies is collected from different tools.

Notably, almost all studies included presented clear research questions and collected data that permitted to clearly address the questions. Overall, these studies exhibited good quality when assessed in terms of collection, analysis, and interpretation. As for studies that used mixed methods, almost all abided by the quality standards for the utilized methods, and both quantitative and qualitative data were consistent. Regarding the quantitative studies included, which were all descriptive, statistical analyses employed were all adequate. In summary, the evidence generated by the systematic review can be deemed valid.

## Conclusion

This scoping review has shown a bidirectional relationship between ON and social media use, influenced by the characteristics of the platform, duration of use, content themes, and individual-level factors. Longitudinal, experimental and

qualitative studies in future research are highly warranted, which will establish causality, help to validate standardized tools assessing both ON and social media use, and provide further insight onto this relationship. Studies must also be done among culturally-diverse sample. The more these dynamics are understood, the better researchers and practitioners in public health can design online and offline environments that support mental health and well-being regarding eating behaviors and social media. This might indicate that public health strategies must work to make social media use beneficial through accessible and supportive messaging.

## Supporting information

**S1 File. Data extraction and quality assessment.** Data extraction of included studies and quality assessment measures adapted from the Mixed Methods Appraisal Tool (MMAT) version 2018.
(XLSX)

**S2 File. Preferred reporting items for systematic reviews and meta-analyses extension for scoping reviews (PRISMA-ScR) checklist.** JBI = Joanna Briggs Institute; PRISMA-ScR = Preferred Reporting Items for Systematic reviews and Meta-Analyses extension for Scoping Reviews. * Where sources of evidence (see second footnote) are compiled from, such as bibliographic databases, social media platforms, and Web sites. † A more inclusive/heterogeneous term used to account for the different types of evidence or data sources (e.g., quantitative and/or qualitative research, expert opinion, and policy documents) that may be eligible in a scoping review as opposed to only studies. This is not to be confused with information sources (see first footnote). ‡ The frameworks by Arksey and O'Malley [6] and Levac and colleagues [7] and the JBI guidance [4,5] refer to the process of data extraction in a scoping review as data charting. § The process of systematically examining research evidence to assess its validity, results, and relevance before using it to inform a decision. This term is used for items 12 and 19 instead of "risk of bias" (which is more applicable to systematic reviews of interventions) to include and acknowledge the various sources of evidence that may be used in a scoping review (e.g., quantitative and/or qualitative research, expert opinion, and policy document). From: Tricco AC, Lillie E, Zarin W, O'Brien KK, Colquhoun H, Levac D, et al. PRISMA Extension for Scoping Reviews (PRISMAScR): Checklist and Explanation. Ann Intern Med. 2018;169:467–473. https://doi.org/10.7326/M18-0850
(DOCX)

## Author contributions

**Conceptualization:** Emmanuelle Awad, Jessica M. Alleva, Carolien Martijn, Rana Rizk.

**Data curation:** Emmanuelle Awad, Celine El Khoury, Nour Chamma.

**Formal analysis:** Emmanuelle Awad, Jessica M. Alleva, Celine El Khoury, Nour Chamma, Carolien Martijn, Rana Rizk.

**Investigation:** Emmanuelle Awad, Celine El Khoury, Nour Chamma.

**Methodology:** Emmanuelle Awad, Jessica M. Alleva, Carolien Martijn, Rana Rizk.

**Project administration:** Emmanuelle Awad.

**Supervision:** Jessica M. Alleva, Carolien Martijn, Rana Rizk.

**Visualization:** Jessica M. Alleva, Carolien Martijn, Rana Rizk.

**Writing – original draft:** Emmanuelle Awad.

**Writing – review & editing:** Emmanuelle Awad, Jessica M. Alleva, Celine El Khoury, Carolien Martijn, Rana Rizk.

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
