## [Decision Letter · Decision Letter 0]

18 Sep 2025

Dear Dr. Awad,

We look forward to receiving your revised manuscript.

Kind regards,

Jenna Scaramanga

Staff Editor

PLOS ONE

Journal Requirements:

“NO authors have competing interests”

3. In this instance it seems there may be acceptable restrictions in place that prevent the public sharing of your minimal data. However, in line with our goal of ensuring long-term data availability to all interested researchers, PLOS’ Data Policy states that authors cannot be the sole named individuals responsible for ensuring data access (http://journals.plos.org/plosone/s/data-availability#loc-acceptable-data-sharing-methods).

Reviewers' comments:

Reviewer's Responses to Questions

**Comments to the Author**

1. Is the manuscript technically sound, and do the data support the conclusions?

Reviewer #1: Yes

2. Has the statistical analysis been performed appropriately and rigorously?

Reviewer #1: Yes

3. Have the authors made all data underlying the findings in their manuscript fully available?

Reviewer #1: Yes

4. Is the manuscript presented in an intelligible fashion and written in standard English?

Reviewer #1: Yes

Reviewer #1: Dear Authors,

The manuscript presents a comprehensive scoping review on the relationship between orthorexia nervosa (ON) and social media. The topic is highly relevant and addresses a growing gap in the health sciences literature. The authors structured the study according to PRISMA-ScR guidelines, registered their protocol on OSF, and performed a wide database search, which demonstrates methodological transparency.

Recommendations

The majority of included studies are cross-sectional, limiting causal inference. While this is noted, the discussion should provide clearer recommendations for future longitudinal and experimental designs.

The limitations of measurement tools (e.g., ORTO-11, ORTO-15) regarding validity and reliability should be emphasized more strongly.

The review is heavily based on Western populations, which reduces cultural diversity and generalizability. This limitation should be highlighted more explicitly in the conclusions.

The results section contains detailed tables; however, additional synthesis (e.g., thematic grouping within tables) would improve readability.

Although the methodology section mentions the possibility of a meta-analysis, the results are purely narrative. If meta-analysis was not feasible, the reasons should be clarified.

Additionally, I recommended articles which should be added in the paper:

Bayram, H. M., Barcın-Guzeldere, H. K., Ede-Cintesun, E., & Çelik, Z. M. (2024). Exploring the interplay between social media addiction, mindful eating, intuitive eating, orthorexia nervosa, and mental health in young adults. North African Journal of Food and Nutrition Research, 8(18), 70-79.

Bayram, H. M. (2024). The role of mindful eating, and intuitive eating on the relationship with orthorexia nervosa in University Students: a cross-sectional study. Revista de Nutrição, 37, e230219.

Demir, H. P., & Bayram, H. M. (2022). Orthorexia nervosa: The relationship with obsessive-compulsive symptoms and eating attitudes among individuals with and without healthcare professionals. Mediterranean Journal of Nutrition and Metabolism, 15(1), 23-33.

**Do you want your identity to be public for this peer review?** For information about this choice, including consent withdrawal, please see our Privacy Policy

Reviewer #1: No

---

## [Author Response · Author response to Decision Letter 1]

21 Oct 2025

Thank you for taking the time to review our manuscript. We greatly appreciate your efforts and consideration. We have adjusted the manuscript to accommodate each recommendation of the reviewer and believe this has led to considerably improvement. Therefore, we want to thank the reviewer for their valuable advice. In the remainder of this letter, we will explain and describe how we have addressed each comment. (The answer to each comment, addressed, can be found below.)

Reviewer #1: Dear Authors,

The manuscript presents a comprehensive scoping review on the relationship between orthorexia nervosa (ON) and social media. The topic is highly relevant and addresses a growing gap in the health sciences literature. The authors structured the study according to PRISMA-ScR guidelines, registered their protocol on OSF, and performed a wide database search, which demonstrates methodological transparency.

Recommendations

1. The majority of included studies are cross-sectional, limiting causal inference. While this is noted, the discussion should provide clearer recommendations for future longitudinal and experimental designs.

Thank you for this valuable suggestion. Specific recommendations were added as suggested (see Page 19 lines 397 to 408):

“The included studies were observational and cross-sectional, indicating the scarcity of experimental studies and limiting the ability to synthesize findings identifying a cause-effect relationship between ON and social media. As previously presented, social media platforms can potentially have a mitigating or distressing effect on ON.(40) Experimental studies can help pinpoint the role of potential moderators such as social media use duration, content and platforms on ON symptomatology. Also, experimental designs where exposure to healthy food-related content on social media, as opposed to other unrelated content, can help determine potential causality. Adding to that, longitudinal studies can be conducted to assess the temporal frame of the relationship between ON and social media, tracking the progression or changes in orthorexic tendencies in parallel with social media use. Also, it can help identify different trajectories and profiles of participants and their characteristics through longitudinal designs.”

2. The limitations of measurement tools (e.g., ORTO-11, ORTO-15) regarding validity and reliability should be emphasized more strongly.

The limitations of the measurement tools were elaborated (see page 24 lines 514 to 521:

“The scales used in the included studies differed, while most used the ORTO-11 and ORTO-15. However, the cut-off point of these two scales was inconsistent across studies.22,27,33 Also, the psychometric quality of different versions of the ORTO, especially of the first ORTO-15 version, were characterized by low internal consistency.72,73 In sum, because studies use different scales to measure orthorexic tendencies and the suboptimal psychometric qualities of some of these scales, the possibility to compare the outcomes across studies is limited. The development and validation of new scales to measure orthorexic tendencies could ensure higher reliability of assessment.”

3. The review is heavily based on Western populations, which reduces cultural diversity and generalizability. This limitation should be highlighted more explicitly in the conclusions.

We have now added this limitation to the “Strengths and Limitations” and “Conclusion” sections. It was also mentioned in the conclusion, respectively (see page 23 lines 498 to 503; page 25, lines 539-540):

“The included studies were diverse in terms of location in the world, cultural groups making up the samples of participants, and language of the published research. Yet, a large number of included studies were conducted on Westernized populations, which shows that more research conducted among different cultural and geographical regions is needed and can help to provide a more comprehensive and holistic picture of the relationships between ON and social media.”

“Studies must also be done among culturally-diverse sample.”

4. The results section contains detailed tables; however, additional synthesis (e.g., thematic grouping within tables) would improve readability.

Table 3 was modified to group the studies in terms of the direction of the association between Orthorexia Nervosa and Social Media. The first column now shows the direction of the relationship between Orthorexia and Social Media in the first column (ON→SM; SM→ON) (see pages 52-60).

“Table 3. Results of Included Studies”

5. Although the methodology section mentions the possibility of a meta-analysis, the results are purely narrative. If meta-analysis was not feasible, the reasons should be clarified.

A justification was added in the methods section (see page 6 line 113 page 7 lines 114-115):

“Given that the screened and included studies were very heterogeneous in terms of characteristics such as study design, populations, and scales used, we determined that a meta-analysis of the data was not appropriate and restricted our synthesis of the data to a narrative review. Adding to that, our main goal was to identify gaps in the literature. For that reason, a narrative scoping review was the best mean to achieve this goal.”

6. Additionally, I recommended articles which should be added in the paper:

Bayram, H. M., Barcın-Guzeldere, H. K., Ede-Cintesun, E., & Çelik, Z. M. (2024). Exploring the interplay between social media addiction, mindful eating, intuitive eating, orthorexia nervosa, and mental health in young adults. North African Journal of Food and Nutrition Research, 8(18), 70-79. (14)

Bayram, H. M. (2024). The role of mindful eating, and intuitive eating on the relationship with orthorexia nervosa in University Students: a cross-sectional study. Revista de Nutrição, 37, e230219. (8)

Demir, H. P., & Bayram, H. M. (2022). Orthorexia nervosa: The relationship with obsessive-compulsive symptoms and eating attitudes among individuals with and without healthcare professionals. Mediterranean Journal of Nutrition and Metabolism, 15(1), 23-33. (10)

Great suggestions. The additional references were added appropriately (see page 4 line 65; page 4 line 67; page 5 line 81).

---

## [Decision Letter · Decision Letter 1]

26 Nov 2025

Dear Dr. Awad,

We look forward to receiving your revised manuscript.

Kind regards,

Michele Fornaro

Academic Editor

PLOS ONE

Journal Requirements:

Reviewers' comments:

Reviewer's Responses to Questions

**Comments to the Author**

Reviewer #1: All comments have been addressed

Reviewer #2: All comments have been addressed

2. Is the manuscript technically sound, and do the data support the conclusions?

Reviewer #1: Yes

Reviewer #2: Yes

3. Has the statistical analysis been performed appropriately and rigorously?

Reviewer #1: Yes

Reviewer #2: N/A

4. Have the authors made all data underlying the findings in their manuscript fully available?

Reviewer #1: Yes

Reviewer #2: Yes

5. Is the manuscript presented in an intelligible fashion and written in standard English?

Reviewer #1: Yes

Reviewer #2: Yes

Reviewer #1: Dear Authors,

All revision suggestions have been implemented. The manuscript is now ready for publication.

Best regards,

Reviewer #2: 1. Abstract

• I encourage you to divide the abstract into subsections: introduction, methods, results, and conclusion.

• Expand the results section of the abstract and provide a more concise synthesis of the methods.

2. Results

• Lines 221–224: I suggest being more quantitative rather than using expressions such as “a few” or “for the most part.”

• Move Table 2 to the Supplementary Materials and add the overall quality rating of each study to Table 1.

• Line 249: Instead of saying “a number of studies,” specify exactly how many studies.

• In the “Risk of Bias Assessment” paragraph, you do not need to highlight the limitations of your study; this should be done in the Discussion section. Furthermore, try to provide quantitative information even if you are conducting a scoping review. Avoid terms like “the majority of” or “most of” and use precise numbers instead.

Given that you assessed 22 quantitative cross-sectional studies, this is a sufficient number to attempt a meta-analytical synthesis of the evidence. By performing a meta-analysis, you could also stratify the findings according to the type of questionnaire used for ON assessment, which would help address the limitation related to inconsistencies between ORTO-11 and ORTO-15.

**Do you want your identity to be public for this peer review?** For information about this choice, including consent withdrawal, please see our Privacy Policy

Reviewer #1: No

Reviewer #2: No

---

## [Author Response · Author response to Decision Letter 2]

16 Dec 2025

Thank you for taking the time to provide your valuable insight. This helps us improve on our manuscript.

Reviewer #2: 1. Abstract

• I encourage you to divide the abstract into subsections: introduction, methods, results, and conclusion.

The abstract was divided as recommended.

• Expand the results section of the abstract and provide a more concise synthesis of the methods.

Thank you for your comment. We modified based on your recommendation.

“Results: After the identification of studies and data extraction, authors assessed the methodological quality of included studies. The characteristics and findings from the studies were narratively synthesized, with a focus on the platform, duration, and content of social media use. A total of 31 studies were identified between 2017 and 2024, which were predominantly cross-sectional and focused on Westernized populations. The results have shown a bidirectional relationship between ON and social media, influenced by the characteristics of the platform (e.g., image-based), duration of use (e.g., longer use), content themes (e.g., diet and fitness-related), and individual-level factors (e.g., limited health literacy, young adulthood and adolescence, and body dissatisfaction).”

Lines 26-31

2. Results

• Lines 221–224: I suggest being more quantitative rather than using expressions such as “a few” or “for the most part.”

These expressions were removed to reduce ambiguity. The results reflect the consensus on the study’s quality.

“The qualitative and mixed-methods studies included in the current review are methodologically solid with minor limitations. Meanwhile, methodological limitations in the included quantitative studies suggest moderate quality. The quality of the included studies is adequate.”

Lines 169-171

• Move Table 2 to the Supplementary Materials and add the overall quality rating of each study to Table 1.

Table 2 was moved to the supplementary material. A column covering “Overall quality rating” was added to Table 1. A brief description was also added to the manuscript:

“In addition, Table 1 covers the overall quality rating of each study on a scale from 1 to 7, with 7 being the best index of quality. This rating was done based on the MMAT Version 18.(20)”

Lines 144-146

• Line 249: Instead of saying “a number of studies,” specify exactly how many studies.

This sentence was changed as recommended:

“Previously, four studies have shown that people who spend a lot of time on social media have higher ON symptoms irrespective of the type of platform.(27),(29),(41),(44).”

Lines 189-190

• In the “Risk of Bias Assessment” paragraph, you do not need to highlight the limitations of your study; this should be done in the Discussion section.

Thank you for this feedback. The risk of bias assessment paragraph now reflects the limitations of the reviewed studies, as opposed to the limitations of our scoping review, which can be found in the limitations section.

“Risk of bias assessment

Out of the reviewed studies, as detailed in Table 3, 15 reported a high risk of bias, driven primarily by significant methodological limitations. Most studies used cross sectional designs, which limits causal inferences from social media to ON. As a result, it is unknown whether social media exposure causes ON symptoms directly, or whether people with ON are more likely to engage with certain types of social media. A second limitation might be reliance on self-report measures that can generate response biases and inaccuracies in reporting ON symptoms and social media behaviors. Yet, it is most reasonable to ask individuals to self-report ON symptoms. In addition, many of the studies used non-representative samples, and were usually not representative of men nor specific cultural groups, thereby limiting the generalizability of the results to broader populations. Furthermore, there is no validated tool to measure ON across studies or social media addiction. ORTO-11 and ORTO-15 were frequently used, but have been criticized for low internal consistency and yielding different psychometric properties among varying cultures.(23, 35) In some studies, new social media use scales were used, including self-constructed items Geise (35), (36, 57), which might require further research to assess validity and reliability.”

Lines 226-237

Furthermore, try to provide quantitative information even if you are conducting a scoping review. Avoid terms like “the majority of” or “most of” and use precise numbers instead.

Excellent observation. Numbers of studies were specified throughout the manuscript for accuracy:

-“A total of 22 studies were quantitative, cross-sectional designs to determine the prevalence of ON symptoms and their associations with social media use.”

Lines 159-160

-“Out of the reviewed studies, as detailed in Table 3, 15 reported a high risk of bias, driven primarily by significant methodological limitations.”

Lines 227-228

-“It is also important to note that 21 of the included studies involved samples of adults only, meaning individuals over the age of 18, with only one study focusing on the relationship between ON and social media exclusively among adolescents aged between 13 and 18.(53)”

Lines 347-349

-“First, 19 the included studies adopted a convenience sampling method without conducting a sample size calculation, constricting the generalizability of the results within the specified population.”

Lines 403-404

Given that you assessed 22 quantitative cross-sectional studies, this is a sufficient number to attempt a meta-analytical synthesis of the evidence. By performing a meta-analysis, you could also stratify the findings according to the type of questionnaire used for ON assessment, which would help address the limitation related to inconsistencies between ORTO-11 and ORTO-15.

We conducted a scoping review about the relationship between Orthorexia Nervosa and social media to evaluate literature gaps and inform future research. Scoping reviews are a type of knowledge synthesis, that follow a systematic approach to map evidence on a topic to identify main concepts, theories, sources, and knowledge gaps. Hence, they differ from systematic reviews and meta-analysis that primordially aim to answer a focused research question. The main aim of our study was to explore the literature relevant to Orthorexia Nervosa and social media by reviewing different definitions, and theories about the relationship found in the literature, as well as pinpoint research gaps. Scoping reviews are exploratory and descriptive in nature, which differs from systematic reviews with meta-analysis. The purpose of our scoping review and the type of data that emerged in answer to our review question are not the type of evidence that lends itself to a meta-analysis, and little value would be gained in performing such an analysis.

Given the difference in objectives and methodological approach (such as presence vs. absence of meta-analysis), scoping reviews should have different essential reporting items from systematic reviews. In our scoping review we adhered to the PRISMA extension for scoping reviews (PRISMA-ScR) where explicitly Item 13: Summary Measures is Not Applicable. This item from the original PRISMA is not applicable for scoping reviews because a meta-analysis is not done (that is, summary measures are not relevant). The same recommendation is echoed in JBI’s Updated methodological guidance for the conduct of scoping reviews.

Relevant references can be found below:

1. Tricco, A. C., Lillie, E., Zarin, W., O'Brien, K. K., Colquhoun, H., Levac, D., ... & Straus, S. E. (2018). PRISMA extension for scoping reviews (PRISMA-ScR): checklist and explanation. Annals of internal medicine, 169(7), 467-473.

2. Rethlefsen, M. L., Kirtley, S., Waffenschmidt, S., Ayala, A. P., Moher, D., Page, M. J., & Koffel, J. B. (2021). PRISMA-S: an extension to the PRISMA statement for reporting literature searches in systematic reviews. Systematic reviews, 10(1), 39.

3. Peters, M. D., Marnie, C., Tricco, A. C., Pollock, D., Munn, Z., Alexander, L., ... & Khalil, H. (2020). Updated methodological guidance for the conduct of scoping reviews. JBI evidence synthesis, 18(10), 2119-2126.

4. Hadie, S. N. H. (2024). ABC of a scoping review: a simplified JBI scoping review guideline. Education in Medicine Journal, 16(2).

---

## [Editor Report · Decision Letter 2]

18 Dec 2025

Orthorexia nervosa and social media: a mixed-methods scoping review using a systematic methodology

PONE-D-25-36486R2

Dear Dr. Awad,

We’re pleased to inform you that your manuscript has been judged scientifically suitable for publication and will be formally accepted for publication once it meets all outstanding technical requirements.

Kind regards,

Michele Fornaro

Academic Editor

PLOS One

Additional Editor Comments (optional):

Thank you for your revision.
---

## [Editor Report · Acceptance letter]

PONE-D-25-36486R2

PLOS One

Dear Dr. Awad,

I'm pleased to inform you that your manuscript has been deemed suitable for publication in PLOS One. Congratulations! Your manuscript is now being handed over to our production team.

Kind regards,

on behalf of

Dr. Michele Fornaro

Academic Editor

PLOS One